# Diff-Privacy: Diffusion-based Face Privacy Protection

## Abstract

Privacy protection has become a top priority due to the widespread collection and misuse of personal data. Anonymization and visual identity information hiding are two important face privacy protection tasks that aim to remove identification characteristics from face images at the human perception level. However, they have a significant difference in that the former aims to prevent the machine from recognizing identity correctly, while the latter needs to ensure the accuracy of machine recognition. Therefore, it is difficult to train a model to complete these two tasks simultaneously. In this paper, we unify the task of anonymization and visual identity information hiding and propose a novel face privacy protection method based on diffusion models, dubbed Diff-Privacy. Specifically, we train our proposed multi-scale image inversion module (MSI) to obtain a set of SDM format conditional embeddings of the original image. Based on the conditional embeddings, we design corresponding embedding scheduling strategies and construct different energy functions during the denoising process to achieve anonymization and visual identity information hiding. Extensive experiments have demonstrated the effectiveness of our proposed framework in protecting facial privacy.

## 1 Introduction

The widespread application of intelligent algorithms and devices brings convenience together with security concerns. Personal images uploaded on social media platforms or captured through intelligent surveillance systems can be collected and misused by threat models, such as illegal snoopers, unauthorized automatic recognition models, and malicious facial manipulation models, thereby posing a significant threat to personal privacy. On the one hand, we are eager to use technology to improve our quality of life (such as video conferencing), but on the other hand, we are unwilling to give up our personal privacy. Consequently, research in the field of privacy protection has garnered significant importance, with a particular emphasis on safeguarding face images that contain substantial amounts of sensitive information.

Recent face privacy protection methods can be broadly divided into two main categories, *i.e.*, anonymization (Neustaedter et al., 2006; Newton et al., 2005; Gross et al., 2009; Hukkelås et al., 2019; Maximov et al., 2020; Cao et al., 2021; Gu et al., 2020; Li et al., 2023) and visual identity information hiding (Ito et al., 2021; Su et al., 2022). Anonymization methods aim to remove identification characteristics from images while retaining essential facial structures to ensure the functionality of face detection algorithms. Crucially, anonymized faces should maintain a realistic appearance while preventing human observers and facial recognition models from recognizing their identities correctly. Unlike anonymization, face images processed by visual identity information hiding methods are unrecognizable to human observers but can be recognized by machines. Regarding application scenarios, the former (anonymization) allows people to share photos with anonymized faces on public social media. The latter (visual identity information hiding) encrypts private images stored in cyberspace, ensuring the accuracy of facial recognition functions while improving security.

However, the above technologies for face privacy protection often specialize in specific types of protection and rely on high-quality facial datasets for training or continuous online iteration of images. Moreover, these technologies often leave noticeable editing traces and possess limited recovery capabilities. Consequently, there is an urgent need to develop a method that can effectively and flexibly achieve face privacy protection for various requirements, mitigating the shortcomings of existing ap-

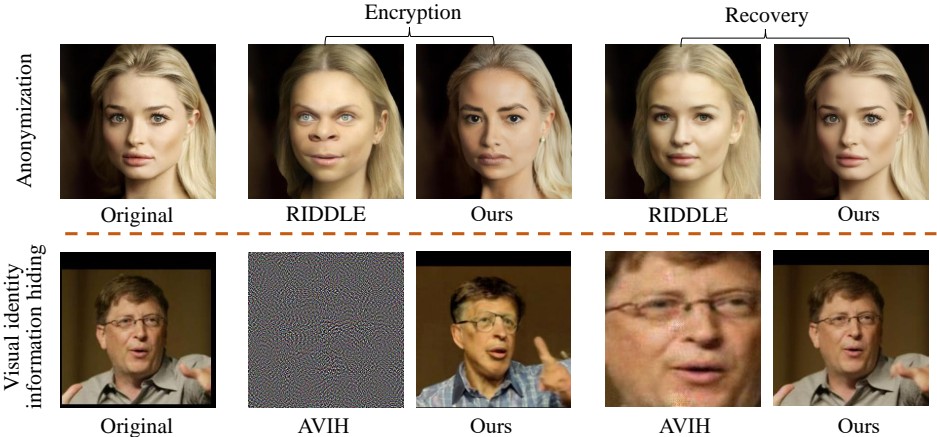

Figure 1: We show the results of our method in anonymization and visual identity information hiding tasks. Compared with existing methods (Li et al., 2023; Su et al., 2022), our method can better generate photo-realism encrypted images and recover the original image.

proaches. Encouraged by the powerful generation prior of diffusion models, we utilize pre-trained diffusion models to promote face privacy protection.

In this paper, we unify the task of anonymization and visual identity information hiding and propose a novel diffusion-based face privacy protection method, dubbed Diff-Privacy. Diff-Privacy enables flexible face privacy protection and ensures identity recovery when needed. As shown in Fig. 1, Diff-Privacy generates highly realistic faces whose identities differ from the original faces during the task of anonymization and visual identity information hiding. Furthermore, Diff-Privacy also exhibits exceptional recovery quality and some other advantages. These include: **Security.** Given an encrypted image, the original image is recovered only if the correct password is provided. **Utility.** The encrypted images generated by our method can still be used for downstream computer vision tasks, such as face detection. **Diversity.** Given an original image, our model can generate a series of encrypted images that are different from each other. **Controllability.** Our method can achieve face privacy protection while identity-independent attributes such as background and posture remain unchanged. We have conducted extensive experiments on publicly available facial datasets to verify the effectiveness of our method.

In summary, we make the following contributions:

- We propose a novel diffusion-based face privacy protection method, which can achieve anonymization or visual information hiding tasks by slightly tweaking some parameters during inference.

- We develop an energy function-based identity guidance module to perform gradient correction on the denoising process to ensure the machine can correctly or incorrectly recognize identity under different privacy protection tasks. We enhance the diversity of anonymized images by maximizing the difference of identities generated under different noises.

- According to the characteristics of the diffusion model that different time steps pay attention to different-level information, we design a multi-scale image inversion module to learn conditional embedding and propose corresponding embedding scheduling strategies to meet different privacy protection requirements.

- Experimental results show that compared with existing methods, our method can significantly change facial identity visually while maintaining its photo-realism, in addition to high-quality recovery results.

## 2 DIFF-PRIVACY

We introduce Diff-Privacy, a diffusion-based face privacy protection method to fulfill the goals we mentioned in Section 1. One notable advantage of Diff-Privacy is its inherent flexibility in achieving face privacy protection to meet diverse requirements. Specifically, the main framework of the

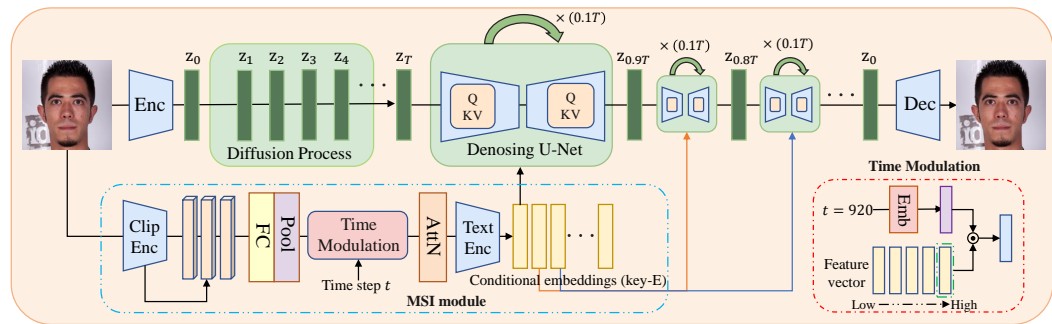

Figure 2: Training process (Stage I) of Diff-Privacy. We apply the SDM as the generative backbone and propose a multi-scale image inversion module. During the training process, the parameters of SDM are fixed. We only optimize our MSI module to extract a set of conditional embeddings.

method can be summarized into the following three stages. **Stage I: Training.** Learning the corresponding conditional embedding of the image in the pre-trained stable diffusion model (SDM) as the key-E. **Stage II: Encryption.** Accomplishing privacy protection through our energy function-based identity guidance module and embedding scheduling strategy during the denoising process and then getting a noised map as the key-I according to DDIM inversion. **Stage III: Recovery.** Performing identity recovery using DDIM sampling based on the acquired key. We will first briefly review some preliminaries in Section 2.1. In Section 2.2, We provide the training process for learning the conditional embedding of the original image. Based on the learned conditional embedding, we design embedding scheduling strategies and an energy function-based identity guidance module to achieve face privacy protection in Section 2.3. Last, we describe the details of how to recover the original face images in Section 2.4.

## 2.1 PRELIMINARIES

Diffusion Models (Ho et al., 2020) are probabilistic models designed to learn a data distribution $p(x)$ by gradually denoising a normally distributed variable, which corresponds to learning the reverse process of a fixed Markov chain of length $T$. In this paper, we apply a pre-trained SDM as the generative backbone. Specifically, the noising process refers to the process of gradually adding Gaussian noise to the initial latent code $z_0$ until the data becomes random noise $z_T$, which is known as a fixed-length Markov chain. An important property of this process is that we can directly sample the noised latent code $z_t$ at any step $t \in \{0, ..., T\}$ based on the original data $z_0$:

$$z_t = \sqrt{\alpha_t} z_0 + \sqrt{1 - \alpha_t} \epsilon, \tag{1}$$

where $\alpha_t = \prod_{i=1}^{t} (1 - \beta_i)$. $\beta_i \in (0, 1), \beta_1 < \beta_2 < ... < \beta_T$. Regarding the reverse process, since we aim to recover the original image accurately, we employ the deterministic DDIM sampling (Song et al., 2020a):

$$z_{t-1} = \sqrt{\frac{\alpha_{t-1}}{\alpha_t}} z_t + \left( \sqrt{\frac{1}{\alpha_{t-1}} - 1} - \sqrt{\frac{1}{\alpha_t} - 1} \right) \cdot \epsilon_\theta \left( z_t, t, C \right), \tag{2}$$

where $C = \phi(P)$ is the embedding of the text condition $P$, $\phi$ is the pre-trained model that maps text condition into a conditional vector. $\epsilon_\theta$ is a time-conditioned UNet equipped with attention mechanism and trained to achieve the objective. The training objectives are as follows:

$$min_\theta E_{z_0, \epsilon \sim \mathcal{N}(0, I), t} \| \epsilon - \epsilon_\theta \left( z_t, t, C \right) \|_2^2. \tag{3}$$

**DDIM inversion.** A simple inversion technique was suggested for the DDIM sampling (Song et al., 2020a), based on the assumption that the ODE process can be reversed in the limit of small steps:

$$z_{t+1} = \sqrt{\frac{\alpha_{t+1}}{\alpha_t}} z_t + \left( \sqrt{\frac{1}{\alpha_{t+1}} - 1} - \sqrt{\frac{1}{\alpha_t} - 1} \right) \cdot \epsilon_\theta \left( z_t, t, C \right). \tag{4}$$

## 2.2 CONDITIONAL EMBEDDING LEARNING

Motivated by text inversion (Gal et al., 2022), we know that within the text embedding space of a stable diffusion, appropriate vectors can be learned to guide the diffusion model in reconstructing a given image. In addition, (Daras & Dimakis, 2022; Zhang et al., 2023a; Balaji et al., 2022) have demonstrated that the generation process of the diffusion model is related to the frequency of the corresponding attribute's signal. Generally, the model tends to generate the overall layout at the initial stage of the denoising process (corresponding to a large time step), the structure and content at the intermediate stage, and the detailed texture at the final stage. Based on these observations, our key insight is whether we can learn a set of embedding vectors of a specific image and utilize them to help achieve privacy protection and image recovery.

We refer to the learnable embedding vectors as "conditional embedding". Our goal is to learn a set of conditional embeddings $\{C\}$ corresponding to the given image. We do not view the diffusion process as a whole but rather as different stages of generation, each stage corresponding to a unique conditional embedding, generating corresponding attributes. Specifically, we divide the 1000 steps of conditioning in SDM into ten stages on average.

$$\{C\} = C^0, C^1..., C^9, \tag{5}$$

where $C^i$ represents the conditional embedding used in the $i^{th}$ stage of generation process.

An instinctive way to obtain this set of embeddings $\{C^i\}$ is to directly optimize it by minimizing the SDM loss of a specific image. However, it is inefficient (Zhang et al., 2023b) and difficult to obtain accurate embeddings without overfitting. Given that our diffusion model employs distinct conditional embeddings at different stages to generate corresponding attributes, we propose the acquisition of conditional embeddings from multi-scale features and the incorporation of temporal information. Thus, we design a multi-scale image inversion (MSI) module to learn conditional embeddings. Specifically, the MSI module utilizes five layers of features from the CLIP image encoder $\tau_\theta$ and maps each layer of features to a vector. After obtaining five vectors, we modulated the vectors through our time modulation module. The time modulation module maps time steps into time embedding and performs point multiplication with corresponding feature vectors, which adaptively adjust the information intensity derived from the features. Subsequently, the MSI module executes attention on embeddings, extracting pivotal information and transmitting it to the text encoder (Radford et al., 2021) for obtaining the final embeddings.

$$\{C\} = MSI(\tau_\theta(x)). \tag{6}$$

We optimize MSI module by minimizing the SDM loss function. The overall training process is illustrated in Fig. 2. To avoid overfitting, we apply a dropout strategy in each cross-attention layer, which is set to $0.05$. Our optimization goal can finally be defined as:

$$L_{SDM} = E_{z,\epsilon\sim\mathcal{N}(0,I),t}\left[\|\epsilon - \epsilon_\theta\left(z_t, t, C_t\right)\|_2^2\right], C_t = C^{t//100}, \tag{7}$$

where $C_t$ denotes the embedding used in time step $t$ and $//$ represents obtaining the integer of quotients in division operations. $z \sim E(x), \epsilon \sim \mathcal{N}(0, I), \epsilon_\theta$ and $\tau_\theta$ are fixed during training.

## 2.3 FACE PRIVACY PROTECTION

To achieve privacy protection, we develop an energy function-based identity guidance module and design embedding scheduling strategies to guide the denoising process. Alg. 1 shows the inference process of Diff-Privacy, which can achieve anonymization and visual identity information hiding.

### 2.3.1 EMBEDDING SCHEDULING STRATEGY

As mentioned in Section 2.2, we learn a set of conditional embeddings of a specific image and apply them to different stages of the denoising process. Generally, diffusion models generate images in the order of "layout → content/structure → texture/style". Based on this characteristic of the diffusion model, we can design different embedding scheduling strategies to meet various privacy protection needs. For example, starting from different initial noises for denoising, we can employ conditional

embedding in the initial stage to uphold the manifold structure of human faces. Subsequently, in the middle and later stages of denoising, we utilize unconditional embedding (default embedding value when the condition is null) to generate diverse appearances and textures of faces. We will introduce our embedding scheduling strategies for different privacy protection tasks later.

### 2.3.2 ENERGY FUNCTION-BASED IDENTITY GUIDANCE MODULE

One pivotal concern in privacy protection is the identification rate of face images by machines. In order to generate images that can be correctly or incorrectly recognized by machines based on different privacy protection requirements, we propose to incorporate some knowledge of face recognition models into the denoising process for guidance. Inspired by (Yu et al., 2023; Kwon & Ye, 2022; Avrahami et al., 2022; Fei et al., 2023), we construct an energy function $\varepsilon$ to perform gradient correction on SDM. According to FreeDoM (Yu et al., 2023), the formula can be written as:

$$\nabla_{x_t} \log p(c|x_t) \propto -\nabla_{x_t} \varepsilon(c, x_t), \tag{8}$$

$$x_{t-1} = x_t - \lambda_t \nabla_{x_t} \varepsilon(c, x_t), \tag{9}$$

where $\nabla_{x_t} \log p(c|x_t)$ is the correction gradient and $c$ is the condition. $x_t$ represents a noisy image. $\lambda_t$ is a scale factor, which can be seen as the learning rate of the correction term. Because we can not find an existing model to measure the distance between noisy data $x_t$ and condition $c$, acquiring clean images $\hat{x}_0$ from the noisy images $x_t$ of the diffusion process is necessary. Referring to Eq. 1, we can estimate the clean latent code $\hat{z}_0$ from the noisy latent code $z_t$ as follows:

$$\hat{z}_0 = \frac{z_t}{\sqrt{\alpha_t}} - \frac{\sqrt{1-\alpha_t}\epsilon_\theta(z_t, t, C_t)}{\sqrt{\alpha_t}}. \tag{10}$$

Subsequently, by employing the pre-trained decoder $D_{ec}$ of SDM, we can derive a clean image $\hat{x}_0 = D_{ec}(\hat{z}_0)$. Accordingly, the energy function can be written as:

$$\varepsilon(c, x_t) \approx D_\theta(x, \hat{x}_0). \tag{11}$$

where $D_\theta$ is a distance measure function and $x$ is the actual condition (original image $x$). Eq. 11 is reasonable because the clean latent code $\hat{z}_0$ has the same trend of change as the noisy latent code $z_t$. As the distance between the clean image $\hat{x}_0$ (decoded from the clean latent code $\hat{z}_0$) and the conditional image decreases, a corresponding reduction occurs in the distance between the noisy image $x_t$ (decoded from the noisy latent code $z_t$) and the conditional image. Then, according to Eq. 2, Eq. 8, Eq. 9, Eq. 10 and Eq.11, the approximated sampling process can be written as:

$$z'_{t-1} = z_{t-1} - \lambda_t \nabla_{\hat{z}_0} D_\theta(x, \hat{x}_0), \tag{12}$$

$$z_{t-1} = \sqrt{\frac{\alpha_{t-1}}{\alpha_t}} z_t + \left( \sqrt{\frac{1}{\alpha_{t-1}} - 1} - \sqrt{\frac{1}{\alpha_t} - 1} \right) \cdot \epsilon_\theta(z_t, t, C_t). \tag{13}$$

### 2.3.3 ANONYMIZATION

We implement anonymization according to the inference process described in Alg.1. Specifically, we first add random noise to the initial latent code $z_0$ to extract the noisy code $z_t$ according to Eq. 1, where $t = S_{ns} * T$. $S_{ns}$ is a scaling factor that mainly controls the noise strength. Then we denoise it according to Eq. 12, where we use the identity dissimilarity loss $L_{Idis}$ and diversity loss $L_{div}$ as distance measurement functions to construct the energy function, i.e., $\varepsilon(c, x_t) \approx D_\theta(x, \hat{x}_0) = L_{Idis} + L_{div}$. The identity dissimilarity loss function ensures that the generated image has a different identity from the original image.

$$L_{Idis} = \sum_{i=1}^{4} Max(\frac{F_\theta(x) \cdot F_\theta(\hat{x}_0^i)}{||F(x)|| \cdot ||F_\theta(\hat{x}_0^i)||}, 0), \tag{14}$$

---

**Algorithm 1** Inference process (Stage II) of Diff-Privacy

---

**Require**: original images $x$, SDM ($\epsilon_\theta$, $D_{ec}$, $E_{nc}$), distance measure function $D_\theta$, a set of conditional embeddings $\{C\}$, total time steps $T$, pre-defined parameters $\alpha_t$, scale factor $\lambda_t$, $S_{ns}$.
**Output**: encrypted image $x_e$, Key-I: $z_T$

1: $z_0 = E_{nc}(x)$
2: Sample $\hat{T} = S_{ns} * T$, $\epsilon \sim \mathcal{N}(0, I)$
3: $z_{\hat{T}} = \sqrt{\alpha_t}z_0 + \sqrt{1 - \alpha_t}\epsilon$                               \\ Noising
4: **for** $t$ from $\hat{T}$ to $1$ **do**
5:    $\hat{x}_0 = D_{ec}(\frac{z_t}{\sqrt{\alpha_t}} - \frac{\sqrt{1-\alpha_t}\epsilon_\theta(z_t,t,C_t)}{\sqrt{\alpha_t}})$,  $C_t = C^{t//100}$
6:    $z_{t-1} = \sqrt{\frac{\alpha_{t-1}}{\alpha_t}}z_t + \left(\sqrt{\frac{1}{\alpha_{t-1}}-1} - \sqrt{\frac{1}{\alpha_t}-1}\right) \cdot \epsilon_\theta(z_t, t, C_t)$
7:    $z'_{t-1} = z_{t-1} - \lambda_t \nabla_{\hat{z}_0} D_\theta(x, \hat{x}_0)$            \\ Denoising with guidance
8: **end for**
9: **for** $t$ from $0$ to $T - 1$ **do**
10:   $z_{t+1} = \sqrt{\frac{\alpha_{t+1}}{\alpha_t}}z_t + \left(\sqrt{\frac{1}{\alpha_{t+1}}-1} - \sqrt{\frac{1}{\alpha_t}-1}\right) \cdot \epsilon_\theta(z_t, t, C_t)$   \\ DDIM inversion
11: **end for**
**Return**: $x_e = D_{ec}(z'_0)$, $z_T$

---

where $F_\theta$ represents a pre-trained face recognition model. Note that in order to enhance the diversity of generated results, we add different noises to the initial latent code $z_0$ and obtained different clean images $\hat{x}_0^i$ based on the Eq. 1, Eq. 13. The detailed process will be introduced below.

Diversity loss is designed to enhance the diversity of anonymized faces. The intuition is to add different noises to the initial latent code (Eq. 1), and the identity of the resulting anonymized face should also be inconsistent. In our experiment, four groups of different noises are added to each initial latent code $z_0$. The diversity loss can be formulated as follows:

$$L_{div} = \sum_{i=1}^{4} \sum_{j=2, j\neq i}^{4} Max\left(\frac{F_\theta(\hat{x}_0^i) \cdot F_\theta(\hat{x}_0^j)}{||F_\theta(\hat{x}_0^i)|| \cdot ||F_\theta(\hat{x}_0^j)||}, 0\right). \tag{15}$$

In addition to utilizing the identity guidance module, we develop an embedding scheduling strategy to implement anonymization more effectively. Specifically, we consider a two-stage embedding scheduling procedure, divided by $\tau \in [0, 1]$. For a denoising process with $T$ steps, one can employ unconditional embedding $UC$ at the first $\tau * T$ steps, and use the conditional embedding $C^{t//100}$ corresponding to each step for the remaining $(1 - \tau) * T$ steps.

$$C_t = \begin{cases} C^{t//100} & t > \tau * T, \\ UC & t \leq \tau * T. \end{cases} \tag{16}$$

This strategy enables us to achieve anonymization while preserving the overall layout of the original image and certain identity-independent attributes (e.g., posture) unchanged. Furthermore, in order to recover the image better, we also generated a noisy latent $z_T$ as key-I through DDIM inversion (Eq. 4) in the process of anonymization, where the conditional embedding $C = C^{t//100}$.

### 2.3.4 VISUAL IDENTITY INFORMATION HIDING

The process of implementing visual identity information hiding is similar to anonymization. Initially, we employ Eq. 1 to introduce noise to the initial latent code $z_0$ and obtain the noisy latent code $z_t$. In this context, $S_{ns}$ is set to a larger value to ensure there is a significant change in face images (e.g., layout and background) from the human observer's perspective. Next, we also obtain the latent code $z_{t-1}$ of step $t - 1$ with guidance through Eq. 12. We use identity similarity loss $L_{is}$ to construct the energy function. Identity similarity loss promotes the generated image that can be correctly recognized by machines, which can be formulated as:

$$L_{is} = 1 - \frac{F_\theta(x) \cdot F_\theta(\hat{x}_0)}{||F_\theta(x)|| \cdot ||F_\theta(\hat{x}_0)||}. \tag{17}$$

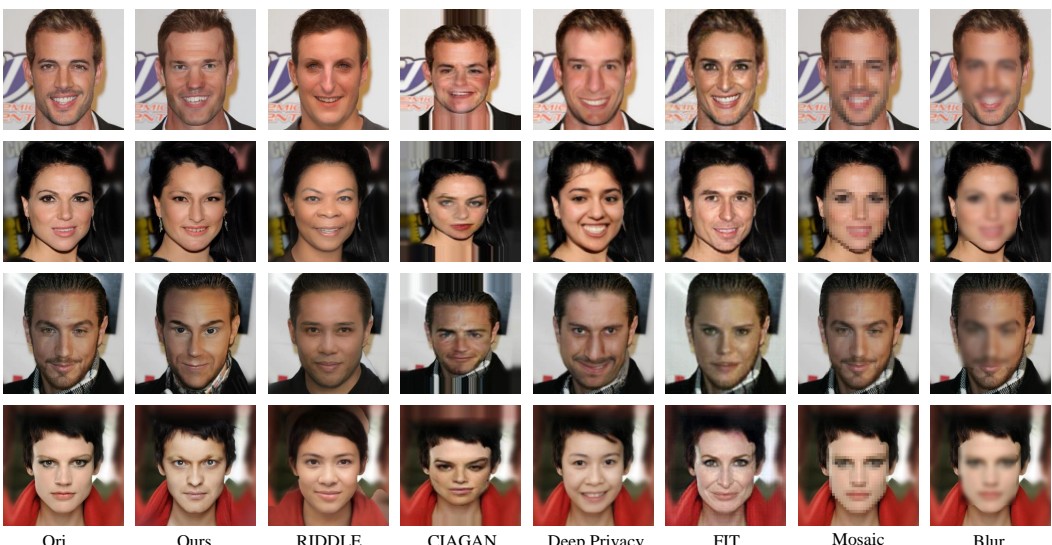

| Ori | Ours | RIDDLE | CIAGAN | Deep Privacy | FIT | Mosaic | Blur |

Figure 3: Qualitative comparison with literature anonymization methods.

Regarding the scheduling strategy for embedding, as we only need to constrain the image to a facial manifold without maintaining some attributes such as posture, we can set $\tau$ in Eq. 16 to a larger value than in the anonymization task. In addition, in order to recover the original image, we also use DDIM Inversion (Eq. 4) to generate a noisy latent code $z_T$ as the key-I.

## 2.4 IDENTITY RECOVERY

In this section, we use the noisy latent code $z_T$ as key-I and a set of conditional embeddings $\{C\}$ as key-E for denoising to recover the original image. Specifically, through Eq. 13, we can gradually denoise the noisy latent code $z_T$ to obtain the recovered latent code $z_r$ and decode it to obtain the recovered image $x_r$, where $C_t = C^{t//100}$.

## 3 EXPERIMENTS AND EVALUATION

In this section, we evaluate the performance of our method in anonymization and visual identity information hiding tasks. The implementation details and more results can be found in the appendix.

## 3.1 ANONYMIZATION

### 3.1.1 DE-IDENTIFICATION

The qualitative comparison is shown in Fig. 3. Blurring and Mosaicing can successfully erase identity information, but photo-realism has been damaged severely. CIAGAN (Maximov et al., 2020) experienced significant distortion when generating anonymized images. FIT (Gu et al., 2020) can generate faces with different identities from the original image. However, its visual quality is unsatisfactory, and the generated faces are unnatural. As shown in the second line of Fig. 3, FIT generates the face of a middle-aged man, yet the skin and

Table 1: Quantitative evaluation for anonymization methods. We calculate the successful protection rate (SR) for de-identification results and the identification rate for recovered results. A higher rate implies better performance.

| Type | Method | Facenet | ArcFace |
|---|---|---|---|
| De-identity↑ | Ours | **0.988** | **1** |
| | RIDDLE | 0.985 | **1** |
| | CIAGAN | 0.849 | 0.9 |
| | FIT | 0.985 | 0.998 |
| | Deep Privacy | 0.923 | 0.933 |

hairstyle more closely resemble those of a woman. RIDDLE (Li et al., 2023) can generate diverse faces, but some important parts of the anonymous faces are unnatural, such as the eyes in the first row of Fig. 3. Deepprivacy (Hukkelås et al., 2019) can somewhat maintain photo-realism but fails to retain the identity-irrelevant attributes such as expressions. Compared with the above methods, our method generates anonymous images with natural facial features and better photo-realism. Moreover, our method can maintain identity-irrelevant attributes.

Quantitatively, we calculate the successful protection rate (SR) of different methods on the CelebA-HQ dataset (Karras et al., 2017). Note that when the distance between the identity embedding of the de-identified image and the source image exceeds the threshold set by the corresponding facial recognition network, protection is considered successful. Here, we use FaceNet and ArcFace for evaluation and choose the threshold of ArcFace as 0.8 and the FaceNet as 1.1 according to (Schroff et al., 2015). Table 1 shows that our method has a higher SR than other anonymization methods, which proves that our method can erase identity information of face images more effectively.

### 3.1.2 IDENTITY RECOVERY

In this section, we compare our method with recoverable anonymization methods FIT (Gu et al., 2020) and RiDDLE (Li et al., 2023) regarding identity recovery performance. We fed both the original and recovered image into the face recognition model to extract identity embedding and subsequently calculate the Cosine similarity between the Identity Embedding (Cos-IE). Moreover, we also use mean square error (MSE), peak signal-to-noise ratio (PSNR), structural similarity (SSIM), and learned perceptual image patch similarity (LPIPS) as metrics. From Table 2, it can be seen that our method outperforms the existing recoverable anonymization methods in terms of identity recovery. In addition, as shown in

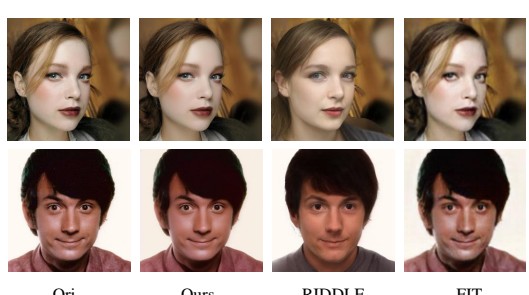

Ori    Ours    RIDDLE    FIT

Figure 4: Visualization comparison of recovered image between different anonymization methods. Zoom in for a better view.

Fig. 4, we can see that compared with FIT and RIDDLE, the recovered image generated by our method is smoother, clearer, and more similar to the original image.

### 3.2 VISUAL IDENTITY INFORMATION HIDING

### 3.2.1 INFORMATION ENCRYPTION

We evaluate our approach according to the testing process proposed by AVIH (Su et al., 2022). We randomly select 12 individuals from the LFW dataset (Huang et al., 2008) as the probe set $Set_p$ and randomly select 10 images for each individual from the probe set as the same-identity verification set $Set_s$. A total of 12878 images from other individuals are used as the different-identity verification set $Set_d$. Then, we encrypt the images of the same-identity verification set $Set_s$. In the evaluation stage, we sequentially take a face image from the probe set as a query and calculate the identity embeddings similarity between the query and images of the same identity in $Set_s$, and the identity embeddings similarity between the query and the images in $Set_d$. If there are images in $Set_d$ with higher similarity than the images in $Set_s$, it is considered that the identification is incorrect.

The results are shown in Table 3. When using FaceNet as the face recognition model, our method has an identification accuracy of three percentage points higher than AVIH and only three percentage points lower than the identification accuracy of original images. Meanwhile, when utilizing the ArcFace face recognition model, our method exhibited comparable performance to AVIH, approaching the identification accuracy achieved with original images. The results can demonstrate the utility of our method on the face recognition models in practical applications. In addition, the qualitative results are shown in Fig. 5. It can be seen that our method encrypts the original image while preserving the facial structure in a photo-realistic manner. A notable advantage over AVIH is that even if a hacker obtains encrypted images, distinguishing whether these images have been encrypted becomes challenging, thus enhancing privacy protection security.

Table 2: Quantitative comparison of recovered results. (F) and (A) represent that we use FaceNet and ArcFace as face recognition models, respectively.

| Method | Anonymization | | | Visual identity information hiding | | |
|---|---|---|---|---|---|---|
| | FIT | RIDDLE | Ours | AVIH (F) | AVIH (A) | Ours |
| MSE ↓ | 0.006 | 0.045 | **0.003** | **0.003** | 0.004 | 0.004 |
| LPIPS↓ | 0.051 | 0.192 | **0.037** | 0.216 | 0.109 | **0.059** |
| SSIM↑ | 0.762 | 0.494 | **0.854** | 0.775 | 0.793 | **0.872** |
| PSNR↑ | 28.693 | 19.489 | **28.900** | **32.306** | 31.369 | 31.913 |
| Cos-IE (F)↑ | 0.896 | 0.774 | **0.956** | 0.926 | 0.87 | **0.929** |
| Cos-IE (A)↑ | 0.88 | 0.709 | **0.932** | **0.919** | 0.726 | 0.905 |

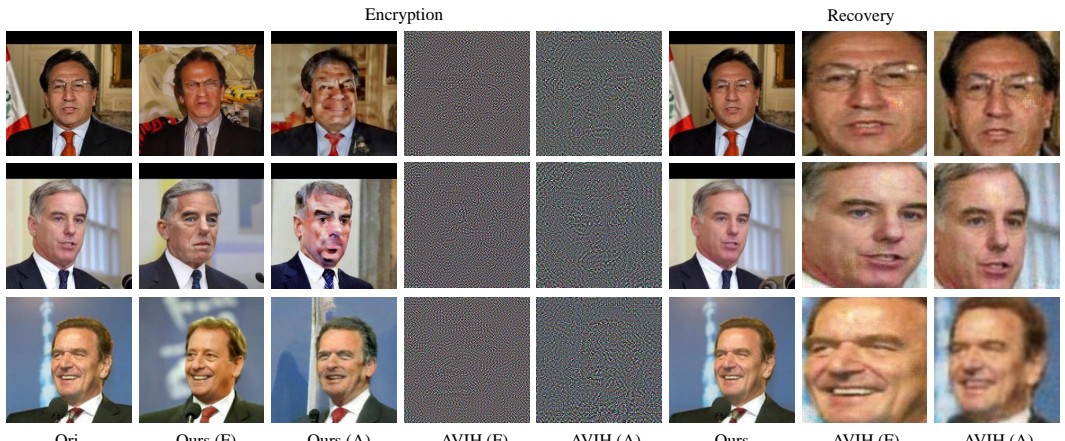

Figure 5: Qualitative comparison with literature visual identity information methods.

### 3.2.2 IDENTITY RECOVERY

In this section, we evaluate the quality of the decrypted image and its similarity to the source image. The results are shown in Fig. 5 and Table 2. Our method achieved comparable performance to AVIH in metrics such as MSE, PSNR, and Cos-IE while surpassing AVIH in LPIPS and SSIM. Moreover, Fig. 5 illustrates that AVIH generates recovered images with artifacts and can only recover the image content of the area where the face is located. In contrast, our method can achieve complete recovery of the original image in a high-quality manner.

Table 3: Face identification accuracy using original images and different encrypted images as validation sets.

| Method | Original | Ours | AVIH |
|---|---|---|---|
| FaceNet | 84.15% | **81.10%** | 78.52% |
| ArcFace | 88.30% | 87.70% | **88.30%** |
| Average | 86.23% | **84.40%** | 83.41% |

## 4 CONCLUSION

In this paper, we unify anonymization and visual identity information hiding tasks and propose a novel diffusion-based face privacy protection method, dubbed Diff-Privacy. It mainly achieves recoverable face privacy protection through three stages. Stage I: Training. Learning a set of conditional embedding of the original image as the key-E through our designed MSI module. Stage II: Encryption. Accomplishing privacy protection through our energy function-based identity guidance module and embedding scheduling strategy during the denoising process and then getting a noised map as the key-I according to DDIM inversion. Stage III: Recovery. Performing identity recovery using DDIM sampling based on the acquired key. Extensive experiments demonstrate our method achieves state-of-the-art results both quantitatively and qualitatively.

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

# A APPENDIX

## A.1 RELATED WORK

### A.1.1 ANONYMIZATION

Anonymization aims to remove identification characteristics from images, rendering them unrecognizable to both human observers and computer vision systems. Existing anonymization methods can be categorized into two main types based on the underlying technology: low-level image processing methods (Boyle et al., 2000; Chen et al., 2007; Neustaedter et al., 2006; Tansuriyavong & Hanaki, 2001; Newton et al., 2005; Gross et al., 2009) and face replacement-based methods (Hukkelås et al., 2019; Gafni et al., 2019; Sun et al., 2018b;a; Maximov et al., 2020; Cao et al., 2021; Gu et al., 2020; Li et al., 2023). The first category anonymization methods, including blurring, mosaicing, pixelization can eliminate identity, but also seriously damage the utility of the original image. Face replacement-based methods (Sun et al., 2018b;a) focus on generating a virtual face to replace the original face. However, the anonymized faces generated by the above methods often have unnatural appearances. Recently, researchers have focused on the development of recoverable anonymization methods (Cao et al., 2021; Gu et al., 2020; Li et al., 2023). Gu et al. (2020) trains a conditional GAN with multi-task learning objectives, which takes the input image and password as conditions and outputs the corresponding anonymization image. Cao et al. (2021) decouples a face image into an attribute vector and identity vector and rotates the identity vector to change identity. Li et al. (2023) projects the original image into the latent space of the pre-trained StyleGAN2 and processes the latent code and password through a lightweight transformer to generate encrypted code. However, these methods often rely on high-quality facial datasets for training and can not achieve satisfactory results in terms of the quality of anonymized and recovered images.

### A.1.2 VISUAL INFORMATION HIDING

Visual information hiding focuses on the human visual perspective, aiming to encrypt the source image so that human observers cannot recognize it. It mainly includes Homomorphic encryption (HE)-based methods (Aono et al., 2017; Liu et al., 2015; Wang et al., 2018) and perceptual encryption (PE)-based methods (Ding et al., 2020; Ito et al., 2021; Sirichotedumrong et al., 2019; Sirichotedumrong & Kiya, 2021). HE-based methods mainly come from Cryptography and are usually unsuitable for deep neural networks (DNNs) containing many nonlinear operations. For PE-based methods, some work (Ding et al., 2020; Sirichotedumrong et al., 2019; Sirichotedumrong & Kiya, 2021) focuses on designing the encrypted domain and directly using the encrypted images to train the model. However, this training strategy has a significant impact on the accuracy of the model. To address this issue, Ito et al. (2021) trains a transformation network to preserve the correct classification results while hiding visual information. However, it generates protected image that can not be recovered to its original form. Inspired by adversarial attack methods, Su et al. (2022) proposes a visual identity information hiding method to protect face privacy protection. However, the generated images are similar to random noise, making it easy for hackers to realize that these images are encrypted. In addition, this online optimization-based method leads to slower generation speed.

### A.1.3 DIFFUSION MODELS

Diffusion-based Generative models (DMs) (Song & Ermon, 2019; Song et al., 2020b; Ho et al., 2020; Sohl-Dickstein et al., 2015; Yang et al., 2022) is a powerful tool for data modeling and generation, which has achieved the first results in density estimation and sample quality. Early work (Ho et al., 2020) relied on markov chains and required many iterations to generate high-quality samples. DDIM (Song et al., 2020a) proposes a deterministic sampling process that greatly reduces the time required to generate samples. Dhariwal & Nichol (2021) proposes to introduce category information into the diffusion model, which can better generate realistic images. However, this method requires to train an additional classifier, resulting in a high training cost. classifier-free diffusion method (Ho & Salimans, 2022) jointly trains the conditional diffusion model and the unconditional Diffusion model, and combines the noise estimation of these two models to achieve the balance between sample quality and diversity. In addition, considering the drawbacks of slow training and inference in

pixel space of the above methods, Rombach et al. (2022) proposes to denoise in the latent space of the pre-trained auto-encoder, significantly reducing the computational requirements.

## A.2 IMPLEMENTATION DETAILS

### A.2.1 PRELIMINARIES

In this section, we provide a concise overview of score function and energy function. Score-based Diffusion Models (SBDMs) (Song & Ermon, 2019; Song et al., 2020b) are a category of diffusion model rooted in score theory. It illuminates that the essence of diffusion models is to estimate the score function $\nabla_{x_t} \log p(x_t)$, where $x_t$ is noisy data. During the sampling process, SBDMs predict $x_{t-1}$ from $x_t$ using the estimated score step by step. Its sampling formula is as follows:

$$x_{t-1} = (1 + \frac{1}{2}\beta_t)x_t + \beta_t \nabla_{x_t} \log p(x_t) + \sqrt{\beta_t}\epsilon, \tag{18}$$

where $\epsilon \in N(0, I)$ is randomly sampled Gaussian noise and $\beta_t \in R$ is a pre-defined parameter. Given sufficient data and model capacity, the score function can be estimated by a score estimator $s(x_t, t)$, that is, $s(x_t, t) \approx \nabla_{x_t} \log p(x_t)$. Nevertheless, the original diffusion process is limited to functioning solely as an unconditional generator, yielding randomly synthesized outcomes. To achieve controllable generation, SDE (Song et al., 2020b)] proposed to control the generated results with a given condition $c$ by modifying the score function as $\nabla_{x_t} \log p(x_t|c)$. Using the Bayesian formula $p(x_t|c) = \frac{p(c|x_t)p(x_t)}{p(c)}$, the conditional score function can be written as two terms:

$$\nabla_{x_t} \log (x_t|c) = \nabla_{x_t} \log p(x_t) + \nabla_{x_t} \log (c|x_t), \tag{19}$$

where the first term $\nabla_{x_t} \log p(x_t)$ can be estimated using the pre-trained unconditional score estimator $s(, t)$ and the second term $\nabla_{x_t} \log (c|x_t)$ is the critical part of constructing conditional diffusion models. The second term can be interpreted as a correction gradient (Yu et al., 2023), irecting $x_t$ towards a hyperplane in the data space where all data conform to the given condition $c$. Following (Yu et al., 2023), we used an energy function (Zhao et al., 2022) to model the correction gradient:

$$p(c|x_t) = \frac{\exp\{-\lambda\varepsilon(c, x_t)\}}{Z}, \tag{20}$$

where $\lambda$ denotes the positive temperature coefficient and $Z > 0$ denotes a normalizing constant, computed as $Z = \int_{c \in c_d} \exp\{-\lambda\varepsilon(c, x_t)\}$ where $c_d$ denotes the domain of the given conditions. $\varepsilon(c, x_t)$ is an energy function that measures the compatibility between the condition $c$ and the noisy image $x_t$. When $x_t$ aligns more closely with $c$, the value of $\varepsilon(c, x_t)$ decreases. If $x_t$ satisfies the constraint of $c$ perfectly, the energy value should be zero. Any function satisfying the above property can serve as a feasible energy function, and we can adjust the coefficient $\lambda$ to obtain $p(c|x_t)$.

Therefore, the correction gradient $\nabla_{x_t} \log (c|x_t)$ can be implemented with the following:

$$\nabla_{x_t} \log (c|x_t) \propto -\nabla_{x_t} \varepsilon(c, x_t) \tag{21}$$

### A.2.2 INFERENCE PIPELINE OF DIFF-PRIVACY

Fig. 6 shows the inference pipeline of Diff-Privacy. We take the anonymization task as an example to provide the inference process. The content enclosed within the black dashed box in the figure signifies its exclusive relevance to the anonymization task. In the inference process, the anonymization task produces four results simultaneously (i.e., batch size=4), while the visual identity information hiding task only produces one result.

### A.2.3 EXPERIMENTAL SETTINGS

We have retained the original hyperparameter selection of SDM, and only the parameters of our proposed MSI module can be trained. On an NVIDIA GeForce RTX3090, the training process for

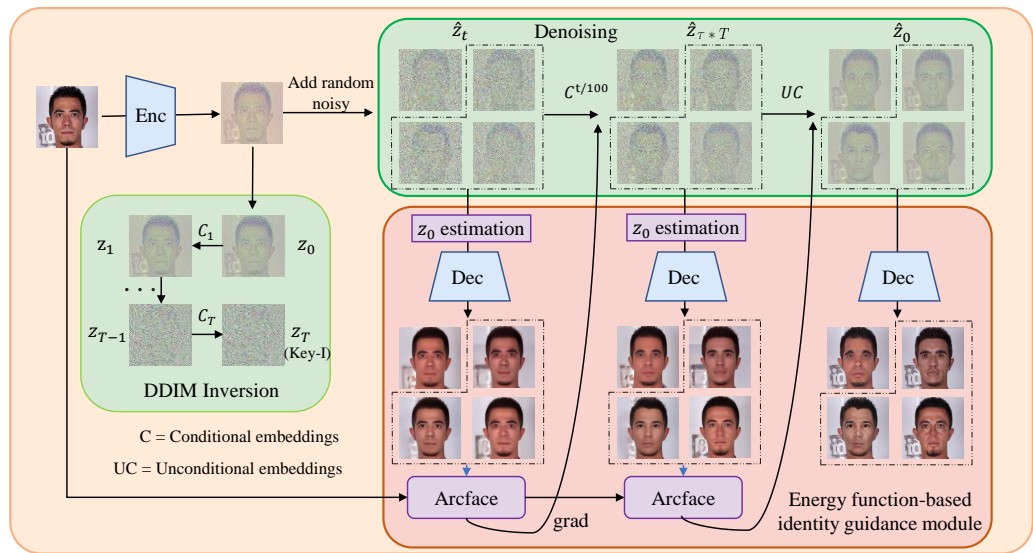

Figure 6: **Inference pipeline (Stage II) of Diff-Privacy.** We take the anonymization task as an example to provide the inference process.

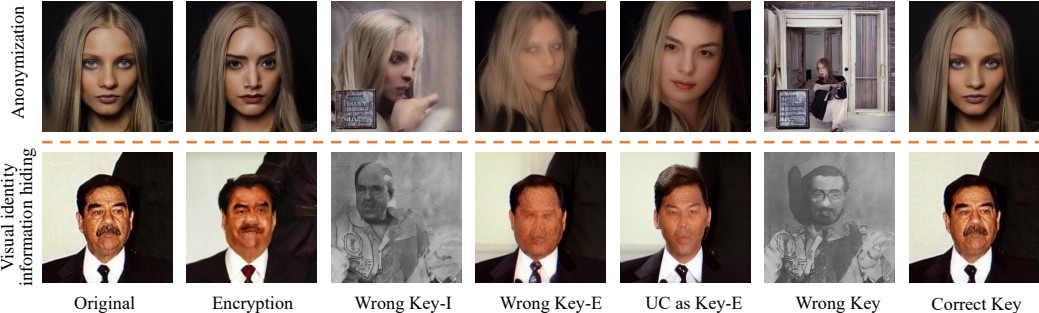

Figure 7: Visualization results of recovered images by using different keys.

each image takes approximately 20 minutes, with a batch size of 1. The basic learning rate is set to 0.001. For evaluation, we use CelebA-HQ Karras et al. (2017) dataset and LFW Huang et al. (2008) dataset to evaluate the effectiveness of Diff-Privacy. In the anonymization task, we perform DDIM sampling with 50 steps, and set a batch size of 4, scaling factor $S_{ns} = 0.6$ and $\tau = 0.4$. For visual identity information hiding task, we perform DDIM sampling with 100 steps, and set a batch size of 4, scaling factor $S_{ns} = 0.8$ and $\tau = 0.6$.

## A.3   MORE RESULTS

In this section, we conduct more comprehensive experiments to evaluate our method. We first evaluate the security of our method in terms of privacy protection. Furthermore, we evaluate the diversity and utility of de-identified images generated in the anonymization task.

### A.3.1   SECURITY

In Section 3, we have verified that our method can achieve privacy protection. Next, we will demonstrate the security of our method. Our method mainly recovers the original image based on key-I and key-E. Now, we obtain the wrong key by randomly sampling or scaling the original key. Then, we use the wrong key to recover the image. Specifically, we set the following variants: wrong key-I, wrong key-E, UC as key-E, wrong key (including key-I and key-E), and correct key. "UC as key-E"

Table 4: Utility evaluation of de-identification results.

| Method | | Ours | RIDDLE | CIAGAN | FIT | DeepPrivacy |
|---|---|---|---|---|---|---|
| Face detection ↑ | MtCNN | **1** | **1** | 1 | **1** | 1 |
| | Dlib | **1** | **1** | 0.957 | **1** | 0.995 |
| Bounding box distance ↓ | MtCNN | 5.834 | 6.45 | 19.975 | **5.3** | 6.467 |
| | Dlib | **3.695** | 6.245 | 18.441 | 3.866 | 5.296 |
| Landmark distance ↓ | MtCNN | 2.848 | 3.478 | 7.326 | **2.398** | 4.298 |
| | Dlib | **2.548** | 3.453 | 9.598 | 2.664 | 4.075 |

Figure 8: Qualitative comparison with literature anonymization methods on diversity.

indicates that we use the correct key-I and use unconditional embedding as key-E to recover the original image. From Fig. 7, it can be seen that as long as there is a key error, it fails to recover the original image, which proves the security of our method. Furthermore, we find that key-I serves as the starting point for denoising, preserving some global information about the original image (such as background and human skin color). On the contrary, key-E contains more detailed information on the face.

### A.3.2 FACE UTILITY

We apply computer vision algorithms on the de-identified images and evaluate the utility of de-identified images on downstream vision tasks. We calculate the face-detection rate between our methods and other anonymization methods on two face-detection models: MtCNN (Zhang et al., 2016) and Dlib Kazemi & Sullivan (2014). The per-pixel distance of facial bounding boxes and 68 facial key points are also calculated. As shown in Table 4, our method achieved the best results in face detection rate and pixel distance calculated by the Dlib model and comparable results as FIT in pixel distance calculated by the MtCNN model. It means that our method can guarantee the consistency of the face region and landmarks better and be used for identity-agnostic computer vision tasks.

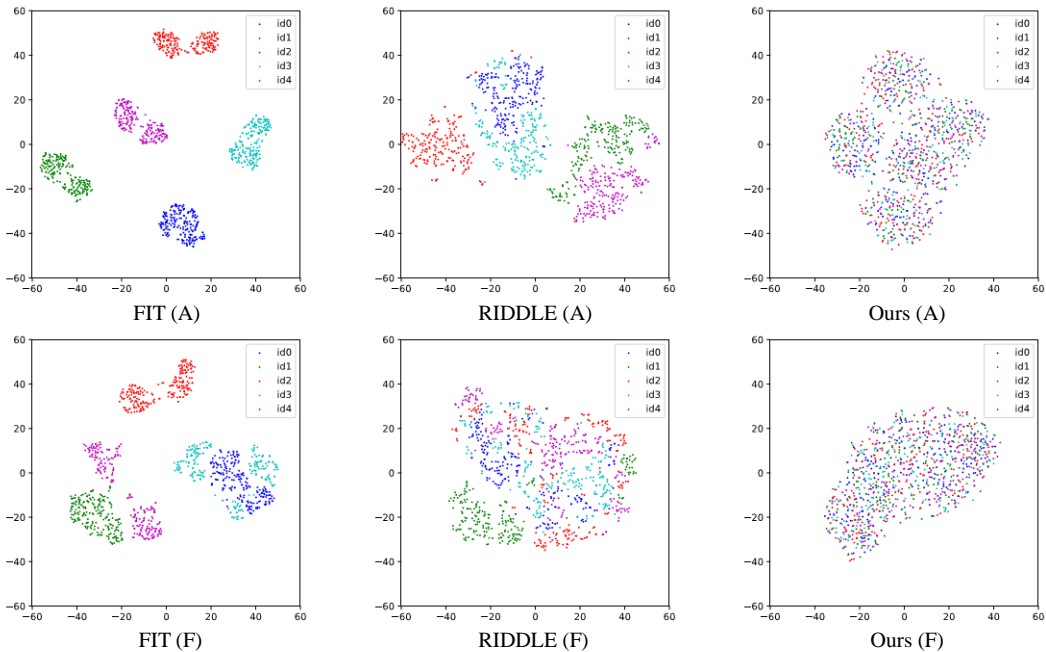

Figure 9: Quantitative comparison with literature anonymization methods on diversity.

Table 5: We calculate the protection success rate in the anonymization task (Task I) and face identification accuracy in the visual identity information hiding task (Task II) to verify the effectiveness of the energy function-based identity guidance module. A higher rate implies better performance.

| Method | Task I | | Task II | |
|---|---|---|---|---|
| | Facenet | ArcFace | FaceNet | ArcFace |
| w/o energy guidance | 14.5% | 99.5% | 0% | 0.06% |
| ours | **98.8%** | **100%** | **81.1%** | **87.7%** |

### A.3.3 DIVERSITY OF IDENTITIES

We compare our method with FIT, RIDDLE, and CIAGAN for the diversity of de-identified images. The results are shown in Fig. 8 and Fig. 9. Although all these methods can generate diverse faces, the faces generated by FIT under different passwords exhibit shared characteristics in local regions. CIAGAN generates faces with low-quality and obvious splicing traces. RIDDLE can obtain diverse faces, but the facial features obtained by encrypting different faces with the same password are similar. In contrast, our method brings fruitful facial features with high-quality for different images. To further demonstrate the diversity of our method, we conduct an identity visualization experiment. For each person, we use 200 different passwords or labels to generate their de-identified faces. Then, we use a face recognition network to extract identity embeddings of de-identified faces and perform dimensionality reduction using t-SNE (Van der Maaten & Hinton, 2008). From Fig. 9, it can be seen that the identity clusters of FIT are relatively tight, and different clusters are spaced far apart on the hyperplane. In contrast, our de-identified faces are more dispersed and occupy most of the area of the hyperplane.

### A.3.4 ABLATION STUDY

In this section, we will evaluate the contribution of each component in our model.

**Energy function-based identity guidance module.** To verify the effectiveness of the energy function-based identity guidance module, we remove the module (represented as **w/o energy guidance**) and conduct experiments on anonymization and visual identity information hiding tasks. From

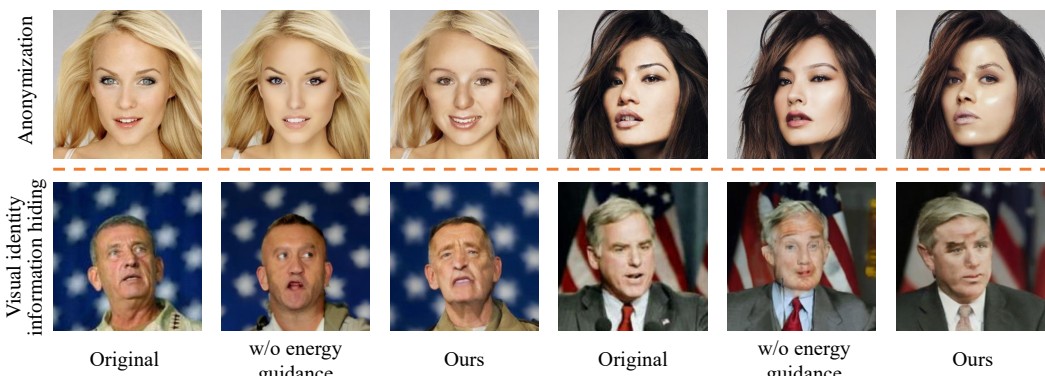

Figure 10: We show two sets of encrypted images in the anonymization and visual identity information hiding tasks. One was generated without an energy function-based identity guidance module, and the other was generated by the full model.

Table 6: The effectiveness of a set of conditional embeddings in identity recovery. Task I represents anonymization and task II represents visual identity information hiding.

| Method | | MSE↓ | LPIPS↓ | SSIM↑ | PSNR↑ | Cos-IE (F)↑ | Cos-IE (A)↑ |
|---|---|---|---|---|---|---|---|
| Task I | Ours-OE | 0.007 | 0.051 | 0.821 | 27.359 | 0.94 | 0.914 |
| | Ours | **0.003** | **0.037** | **0.854** | **28.900** | **0.956** | **0.932** |
| Task II | Ours-OE | 0.005 | 0.080 | 0.831 | 30.237 | 0.913 | 0.856 |
| | Ours | **0.004** | **0.059** | **0.872** | **31.913** | **0.929** | **0.905** |

Fig. 10 and Table 5, it can be seen that removing the energy function-based identity guidance module in the anonymization task can also generate faces with different appearances, but their identities are relatively similar to the original image. In addition, regarding machine perception, removing the energy function-based identity guidance module leads to a decrease in the protection success rate (SR), especially when using FaceNet as the face recognition model. When removing the energy function-based identity guidance module to achieve the visual identity information hiding task, the generated face has significant visual differences from the original face. However, the identification accuracy of using these generated faces as the validation set is almost zero, which demonstrates that the generated face cannot replace the original image to complete face recognition.

**A set of conditional embeddings.** In our method, we design an MSI module to obtain a set of conditional embeddings of the original image. Now, we conduct experiments to verify the advantages of a set of conditional embeddings over one conditional embedding. Specifically, we remove the multi-layer features and time modulation modules from the MSI module and only use the last layer of features encoded by CLIP to obtain one conditional embedding, represented as **Ours-OE**. Fig. 11 shows the de-identified results of using one embedding and a set of conditional embeddings in anonymization tasks. It can be seen that using a set of conditional embeddings has better decoupling and editability, and different levels of privacy protection can be achieved by using different embedding scheduling strategies. On the contrary, using one conditional embedding has poor editability and generates unsatisfactory de-identified faces. In addition, we also conducted experiments on the impact of one embedding and a set of embeddings on identity recovery. The results are shown in Table. 6. From the table, it can be seen that using a set of embeddings for identity recovery outperforms using a single embedding in both pixel-level and perceptual-level metrics. The above experiment demonstrates the effectiveness of employing our MSI module to acquire a set of conditional embeddings for both encryption and recovery processes.

**Embedding scheduling strategy.** We have designed corresponding embedding scheduling strategies for privacy protection tasks with different requirements. Next, we conduct experiments to verify the effectiveness of these strategies and provide the impact of different strategies on the generated results. Specifically, we conduct experimental verification by changing $\tau$ in the embedding scheduling strategy. The results are shown in Fig. 11 and Fig. 12. From the first line of Fig. 11, it can be

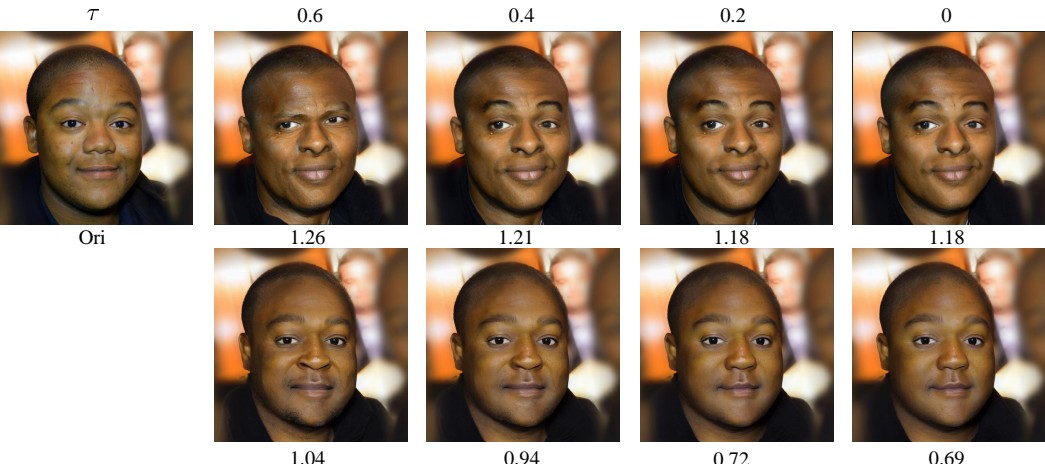

Figure 11: **Effect of different conditional embedding in our method.** The first line of the image is generated using a set of conditional embeddings, while the second line is generated using one conditional embedding. The images in each row, arranged from left to right, represent the outcomes produced by the different $\tau$ of the embedding scheduling strategy. The number below the image represents the distance between the identity embedding of the de-identified face and the identity embedding of the original face.

seen that as $\tau$ decreases, the similarity between the de-identified face and the original face gradually increases, and the identity embedding distance between the de-identified face and the original face decreases. When $\tau = 0.4$, it ensures the effectiveness of anonymization while maintaining irrelevant attributes such as posture and facial expressions unchanged. In addition, it can be seen from Fig. 12 that in the visual identity information hiding task, when $\tau \in [0.4, 0.6]$, the distance between the encrypted face and the original face in the identity embedding space is the smallest. In the visual identity information hiding task, in order to ensure significant changes in the identity of encrypted images from the perspective of human observers, we chose the time embedding scheduling strategy with $\tau = 0.6$.

**Noise strength.** We change the scaling factor $S_{ns}$ to explore the impact of different noise strengths on the generated results. As shown in Fig. 13, when the noise strength weakens (i.e., the scaling factor $S_{ns}$ decreases), the generated results gradually resemble the original image. In the first row of the figure (anonymization task), it can be seen that when $S_{ns} = 0.6$, privacy protection is effectively achieved while ensuring that identity-independent attributes such as posture and facial expressions remain unchanged. In the visual identity information hiding task (the second line in the figure), when $S_{ns} = 0.8$, it maximizes the difference observed by the naked eye while ensuring the correct recognition of identity by the machine.

**Diversity loss function.** In the anonymization task, in order to enhance the diversity of de-identified images, we design a diversity loss function. Now, we remove the diversity loss function and conduct experiments to verify its effectiveness. This variant is called **w/o diversity loss**. We conduct identity visualization experiments as described in Section A.3.3. The results are shown in Fig. 14. With diversity loss function, the generated de-identified faces are more scattered and occupy most of the area of the hyperplane.

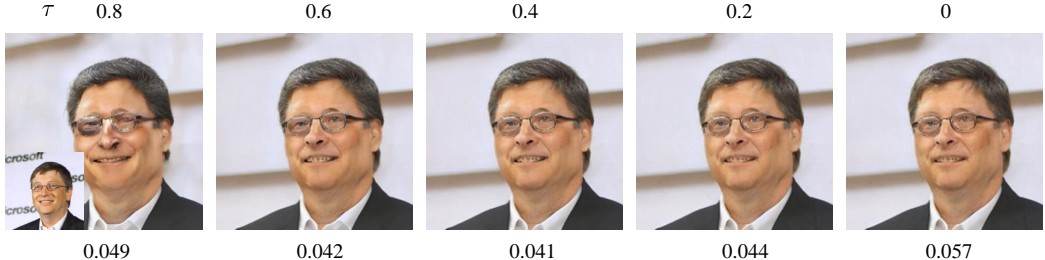

Figure 12: **Visualization of generated results of visual identity information hiding tasks under different embedding scheduling strategies.** The number below the image represents the identity embedding distance between the encrypted face and the original face.

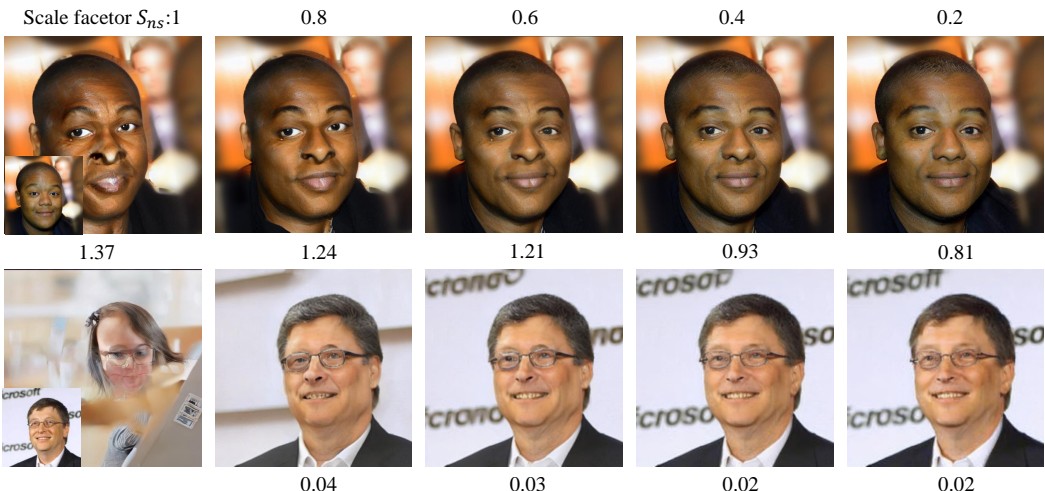

Figure 13: **Effect of different noise strength in our method.** The number below the image represents the distance between the identity embedding of the encrypted face and the identity embedding of the original face. Because the image in the second row and first column can not detect a face, the distance between it and the original face in the identity embedding space is not given.

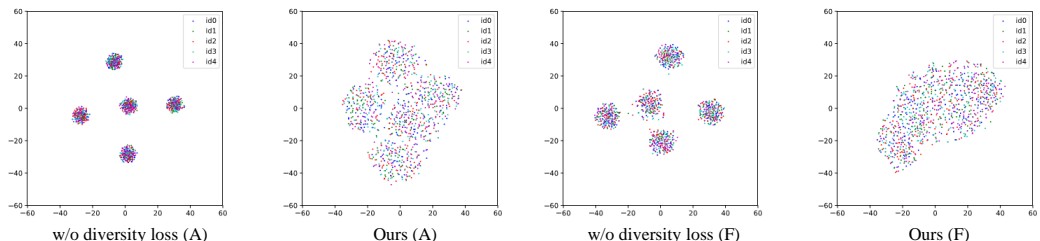

Figure 14: **The effectiveness of diversity loss function.** We remove the diversity loss and conducted experiments to verify its effectiveness.

