# OpenReview forum: "Diff-Privacy: Diffusion-based Face Privacy Protection"
_ICLR.cc/2024/Conference — Submitted to ICLR 2024_

### Official Review · Reviewer_J6vY · 2023-10-28

**Soundness:** 4 excellent
**Presentation:** 3 good
**Contribution:** 4 excellent
**Rating:** 8
**Confidence:** 4

**Summary:**

This paper unifies the task of anonymization and visual identity information hiding and proposes a novel diffusion-based face privacy protection method. Specifically, it learns a set of SDM format conditional embeddings through the MSI module. Then, the authors designed corresponding embedding scheduling strategies and energy functions to guide the denoising process to achieve different privacy protection tasks.

**Strengths:**

1. This study analyzes the similarities and differences between anonymization and visual identity information hiding tasks and proposes the first work that can simultaneously achieve these two tasks.
2. The authors utilize the powerful generation prior of the diffusion model and design reasonable guidance to help SDM complete privacy protection and identity recovery. It is a novel privacy protection framework with unique password and identity recovery patterns.
3. This paper is well organized and written in general. Most of the claims are supported by ample experimental analysis.

**Weaknesses:**

1. The author learns a set of conditional embeddings through a few images. When using the LFW dataset for training, do different images under the same identity need to be trained separately or can they be trained together to obtain a set of conditional embeddings for encryption and decryption?
2. In Sec.2.2, the description of obtaining conditional embeddings seems unclear. Does each time step correspond to a unique time embedding?

**Questions:**

Please see the weaknesses.

---

> ### Author Response · Authors · 2023-11-16
> **Response to Reviewer J6vY**
>
> Thank you for your comments and feedback. We address your concerns here.
>
> >Q1: The author learns a set of conditional embeddings through a few images. When using the LFW dataset for training, do different images under the same identity need to be trained separately or can they be trained together to obtain a set of conditional embeddings for encryption and decryption?
>
> A1: Thank you for the opportunity to clarify. Our MSI module learns conditional embedding through a few images. When the training set is FFHQ or CelebA-HQ, each person has only one image, so we use a single image to learn conditional embedding. When the training set is an image from LFW, we use all images of the person to learn conditional embedding.
>
> >Q2: In Sec.2.2, the description of obtaining conditional embeddings seems unclear. Does each time step correspond to a unique time embedding?
>
> A2: Thank you for the opportunity to clarify. According to previous work [1,2,3], we know that diffusion models focus on different-level information at different time steps. Generally, the model tends to generate the overall layout at the initial stage of the denoising process (corresponding to a large time step), the structure and content at the intermediate stage, and the detailed texture at the final stage. Therefore, we designed an MSI module to learn a set of conditional embeddings as conditions for different periods in the diffusion model. Specifically, we divide the generation process (1000 time steps) into ten time periods, so the output dimension of the time embedding layer is also 10, with each time embedding corresponding to a period.
>
> **Reference**
>
> [1] Wang, Jianyi, et al. "Exploiting Diffusion Prior for Real-World Image Super-Resolution." arXiv preprint arXiv:2305.07015 (2023).
>
> [2] Zhang, Yuxin, et al. "ProSpect: Expanded Conditioning for the Personalization of Attribute-aware Image Generation." arXiv preprint arXiv:2305.16225 (2023).
>
> [3] Balaji, Yogesh, et al. "ediffi: Text-to-image diffusion models with an ensemble of expert denoisers." arXiv preprint arXiv:2211.01324 (2022).

---

### Official Review · Reviewer_pv71 · 2023-10-30

**Soundness:** 4 excellent
**Presentation:** 2 fair
**Contribution:** 3 good
**Rating:** 6
**Confidence:** 4

**Summary:**

Overall this is an interesting paper that proposes a novel diffusion-based method for face privacy protection. The method is flexible and can achieve both anonymization and visual identity information hiding. The results demonstrate state-of-the-art performance.

**Strengths:**

See the summary in detail.

**Weaknesses:**

1. The network architecture shown in Fig.2 is not clear. As mentioned by the authors, the framework of the proposed method can be divided into three stages, which are not presented in Fig.2. Also, key-E, key-I, and the proposed energy function-based identity guidance module are not clearly shown in Fig.2.
2. The energy function in Section 2.3.2 is introduced briefly - more details are needed on the formulation and how it enables identity guidance. In addition, the definition of $\varepsilon$ which indicates energy function is not clearly defined.
3. In the Require part of Alg.1, "scheduling strategy for embedding $C_{T}$" is confusing as the authors define $C_{T}$ as embedding used in time step t before.
4. The writing should be improved. The manuscript is hard to read and there are occasional grammatical issues and unclear/repetitive statements.

**Questions:**

See the summary in detail.

---

> ### Author Response · Authors · 2023-11-16
> **Part 1 of Response to Reviewer pv71**
>
> Thank you for your comments and feedback. We address your concerns here.
>
> >Q2: The energy function in Section 2.3.2 is introduced briefly - more details are needed on the formulation and how it enables identity guidance. In addition, the definition of $\epsilon$ which indicates energy function is not clearly defined.
>
> A2: Thank you for your suggestion. We have provided more details about the formula below and added it to the revised manuscript.
>
> Firstly, we provide a concise overview of Score-based Diffusion Models. Score-based Diffusion Models (SBDMs) [1,2] are a category of diffusion model rooted in score theory. It illuminates that the essence of diffusion models is to estimate the score function $\nabla_{x_t}\log_{}{p(x_t)}$, where $x_t$ is noisy data. During the sampling process, SBDMs predict $x_{t−1}$ from $x_t$ using the estimated score step by step. Its sampling formula is as follows:
>
> $x_{t-1} = (1+\frac{1}{2}\beta_t)x_t+ \beta_t \nabla_{x_t}\log_{}{p(x_t)}+\sqrt \beta_t \epsilon, $
>
> where $\epsilon \in N(0,I)$ is randomly sampled Gaussian noise and $\beta_t \in R$ is a pre-defined parameter. Given sufficient data and model capacity, the score function can be estimated by a score estimator $s(x_t, t)$, that is, $s(x_t, t) \approx \nabla_{x_t}\log_{}{p(x_t)}$. Nevertheless, the original diffusion process is limited to functioning solely as an unconditional generator, yielding randomly synthesized outcomes. To achieve controllable generation, SDE [2] proposed to control the generated results with a given condition $c$ by modifying the score function as $\nabla_{x_t}\log_{}{p(x_t|c)}$. Using the Bayesian formula $p(x_t|c) = \frac{p(c|x_t)p(x_t)}{p(c)}$, the conditional score function can be written as two terms:
>
> $\nabla_{x_t}\log_{}{(x_t|c)} = \nabla_{x_t}\log_{}{p(x_t)}+ \nabla_{x_t}\log_{}{(c|x_t)},$
>
> where the first term $\nabla_{x_t}\log_{}{p(x_t)}$ can be estimated using the pre-trained unconditional score estimator $s(·, t)$ and the second term $\nabla_{x_t}\log_{}{(c|x_t)}$ is the critical part of constructing conditional diffusion models. The second term can be interpreted as a **correction gradient** [3], directing $x_t$ towards a hyperplane in the data space where all data conform to the given condition $c$. Following [3], we used an energy function [4,5] to model the correction gradient:
>
> $p(c|x_t) = \frac{\exp\left\\{-\lambda \varepsilon(c,x_t) \right\\}}{Z},$
>
> where $\lambda$ denotes the positive temperature coefficient and $Z>0$ denotes a normalizing constant, computed as $Z = \int_{c \in c_d} \exp\left\\{− \lambda \varepsilon (c, x_t) \right\\}$ where $c_d$ denotes the domain of the given conditions. $\varepsilon (c, x_t)$ is an **energy function** that measures the compatibility between the condition $c$ and the noisy image $x_t$. When $x_t$ aligns more closely with $c$, the value of $\varepsilon (c, x_t)$ decreases. If $x_t$ satisfies the constraint of $c$ perfectly, the energy value should be zero. Any function satisfying the above property can serve as a feasible energy function, and we can adjust the coefficient $\lambda$ to obtain $p(c|x_t)$.
>
> Therefore, the correction gradient $\nabla_{x_t}\log_{}{(c|x_t)}$ can be implemented with the following:
>
> $\nabla_{x_t}\log_{}{(c|x_t)} \propto - \nabla_{x_t}\varepsilon(c,x_t).$
>
> Based on the above theories and formulas, classifier guidance methods [4,6,7,8] trained a classifier by using noisy data to calculate the correction gradient for conditional guidance. Unlike the classifier guidance approach, we aim to use pre-trained models to calculate the correction gradient. Therefore, we need to estimate clean data $\hat x_0$ from noisy data $x_t$. we use Eq.10 to estimate clean latent codes $\hat z_0$ from noisy latent codes $z_t$ and decode $\hat z_0$ to obtain clean image $\hat x_0$. Once we have a clean image $\hat x_0$, we can use existing facial recognition models to calculate the distance between the clean image and condition and construct an energy function.
>
> $\varepsilon(c,x_t) \approx D_\theta(\hat x_0, x),$
>
> where $D_\theta$ is a distance measure function and $x$ is the actual condition (original image $x$).
>
> Then, we can perform gradient correction on the DDIM sampling process according to the above formula.
>
> $z_{t-1}= \sqrt{\frac{\alpha_{t-1}}{\alpha_t}}z_t + \left(\sqrt{\frac{1}{\alpha_{t-1}}-1}-\sqrt{\frac{1}{\alpha_t}-1}\right)\cdot \epsilon_\theta\left(z_t,t,C_t\right),$
>
> $z_{t-1}' = z_{t-1} - \lambda_t \nabla_{\hat{z}_0} D\_\theta(x,\hat{x}_0),$
>
> where $z_{t-1}$ is obtained through the original ddim sampling (Eq.2). $z_{t-1}'$ is the result obtained after our gradient correction and $\lambda_t$ is a scale factor.

---

> > ### Author Response · Authors · 2023-11-16
> > **Part 2 of Response to Reviewer pv71**
> >
> > >Q1: The network architecture shown in Fig.2 is not clear. As mentioned by the authors, the framework of the proposed method can be divided into three stages, which are not presented in Fig.2. Also, key-E, key-I, and the proposed energy function-based identity guidance module are not clearly shown in Fig.2.
> >
> > A1: Thank you for the opportunity to clarify. As mentioned in our paper, the proposed method can be divided into three stages. However, Fig.2 corresponds only to the first stage of our method, demonstrating the process of learning conditional embedding. The second stage of our method is shown in Fig.6, which shows the process of generating encrypted images. The third stage aims to recover images using DDIM sampling (Eq.13) based on key-I and key-E. To enhance the readability and comprehension of our manuscript, we have furnished a more explicit definition of these three stages in the revised manuscript. Additionally, we have incorporated annotations for key-E and key-I in Fig.2 and Fig.6, addressing your comments for better clarity.
> >
> > >Q3 and Q4: In the Require part of Alg.1, "scheduling strategy for embedding
> > $C_t$" is confusing as the authors define as embedding used in time step t before.
> > The writing should be improved. The manuscript is hard to read and there are occasional grammatical issues and unclear/repetitive statements.
> >
> > A: Thank you for your suggestion. Algorithm 1 has been revised to accurately depict the inference process of our method. Additionally, we have made our best efforts to revise our manuscript to make it easy to read and understand.
> >
> > **Reference**
> >
> > [1] Song, Yang, and Stefano Ermon. "Generative modeling by estimating gradients of the data distribution." Advances in neural information processing systems 32 (2019).
> >
> > [2] Song, Yang, et al. "Score-based generative modeling through stochastic differential equations." arXiv preprint arXiv:2011.13456 (2020).
> >
> > [3] Yu, Jiwen, et al. "Freedom: Training-free energy-guided conditional diffusion model." arXiv preprint arXiv:2303.09833 (2023).
> >
> > [4] Zhao, Min, et al. "Egsde: Unpaired image-to-image translation via energy-guided stochastic differential equations." Advances in Neural Information Processing Systems 35 (2022): 3609-3623.
> >
> > [5] LeCun, Yann, et al. "A tutorial on energy-based learning." Predicting structured data 1.0 (2006).
> >
> > [6] Dhariwal, Prafulla, and Alexander Nichol. "Diffusion models beat gans on image synthesis." Advances in neural information processing systems 34 (2021): 8780-8794.
> >
> > [7] Liu, Xihui, et al. "More control for free! image synthesis with semantic diffusion guidance." Proceedings of the IEEE/CVF Winter Conference on Applications of Computer Vision. 2023.
> >
> > [8] Nichol, Alex, et al. "Glide: Towards photorealistic image generation and editing with text-guided diffusion models." arXiv preprint arXiv:2112.10741 (2021).

---

> > > ### Comment · Reviewer_pv71 · 2023-11-21
> > >
> > > Thank you for your response. The proposed method is solid and I prefer to raise my rating from 5 to 6.

---

> ### Author Response · Authors · 2023-11-22
> **Thank you for your response.**
>
> Thank you for carefully reviewing the discussions and deciding to raise the score. We are glad that we can modify the manuscript with your suggestion to make it easier to read and understand.

---

### Official Review · Reviewer_4Vnn · 2023-11-05

**Soundness:** 3 good
**Presentation:** 3 good
**Contribution:** 3 good
**Rating:** 8
**Confidence:** 4

**Summary:**

This paper develops a new paradigm for facial privacy protection based on the diffusion model. Once trained, this model can flexibly implement different facial privacy protection tasks during inference, such as anonymization and visual identity information hiding. Qualitative and quantitative experiments have demonstrated the effectiveness of the proposed method.

**Strengths:**

Strength:
++ The authors innovatively propose a new paradigm for facial privacy protection based on diffusion models. Quantitative and qualitative experiments have demonstrated the effectiveness of the proposed paradigm.
++ The authors propose an MSI module to learn a set of SDM formats conditional embeddings of the original image and demonstrate that the embeddings extracted by this module have better editability and decoupling.
++ The authors specially design an embedded scheduling strategy and an energy-based identity guidance module to guide the diffusion model, which makes the diffusion model effectively meet the needs of facial privacy protection tasks at the human perception level and machine perception level.

**Weaknesses:**

Weakness:
-- Although the author has significantly reduced the training cost of the model (including the need for high-quality facial datasets), the diffusion-based method for inference is still inefficient compared to the GAN-based method.
-- I noticed that the author conducted an experiment on the security of keys. I think more interesting experiments can be conducted on key-I and key-E to explore their role in image generation.

**Questions:**

Please find the weaknesses.

---

> ### Author Response · Authors · 2023-11-16
> **Response to Reviewer 4Vnn**
>
> Thank you for your comments and feedback. We address your concerns here.
>
> >Q1: Although the author has significantly reduced the training cost of the model (including the need for high-quality facial datasets), the diffusion-based method for inference is still inefficient compared to the GAN-based method.
>
> A1: Thank you for pointing out this issue. Our method inherits the diffusion model's intrinsic drawbacks, notably its sluggish inference speed. Nevertheless, noteworthy strides have been made in expediting sampling by constructing specialized samplers. This development aligns with our objectives, and we can leverage these advancements to enhance the sampling speed of our method in future implementations.
>
> >Q2: I noticed that the author conducted an experiment on the security of keys. I think more interesting experiments can be conducted on key-I and key-E to explore their role in image generation.
>
> A2: Thank you for your suggestion. We have followed your advice and conducted more experiments on the role of keys (as shown in Fig.7). While ensuring the correctness of key-I, we used the unconditional embedding of the diffusion model as the key-E for image recovery. The results indicate that the image can not be recovered correctly as long as there is a password error. Furthermore, key-I serves as the starting point for denoising, preserving some global information about the original image (such as background and human skin color). On the contrary, key-E contains more detailed information on the face.

---

> > ### Comment · Reviewer_4Vnn · 2023-12-02
> >
> > I reviewed the papers cited by the author on anonymization and visual information hiding tasks and found that there is no clear definition of threat models and assumptions in these papers. If a definition is indeed necessary, the author's proposed definition appears reasonable and aligns well with the task setting.

---

### Official Review · Reviewer_jBnK · 2023-11-06

**Soundness:** 2 fair
**Presentation:** 2 fair
**Contribution:** 2 fair
**Rating:** 5
**Confidence:** 3

**Summary:**

The paper proposes a face privacy-preservation method based on diffusion models that claims to achieve both anonymization and visual information hiding within a unified framework.

**Strengths:**

1) The paper addresses an important problem of enhancing privacy for face images shared on the Internet.
2) The proposed solution appears to be somewhat novel and feasible, but it is hard to judge because sufficient details have not been provided.

**Weaknesses:**

1) The paper is extremely hard to read and understand. A number of mathematical notations and terms (e.g., key-E, conditional embedding, etc.) are  not defined clearly, which makes the proposed approach hard to follow.

2) First and foremost, the most basic requirement in a security/privacy paper is stating the threat model and assumptions explicitly. What information is being protected and from whom? What are the capabilities of the adversary? In this paper, no clear threat model has been presented. For example, the goals of anonymization and visual information hiding appear to be quite different. Is the proposed framework designed to meet both these objectives simultaneously? Or is the paper trying to propose a common framework that can achieve either one of these objectives at a time by slighting tweaking some parameter/loss?

3) For anonymization (de-identification), it is not sufficient to show that "anonymized" faces cannot be recognized by a standard face recognition system. It is necessary to prove that the face cannot be "deanonymized" by a malicious adversary. In the proposed approach, it appears that complete recovery is possible provided the so-called "keys" are available. This makes the privacy dependent on the confidentiality of the "key". Even in the absence of the "key", it may be possible for an adversary to learn the inverse mapping between the original and anonymized faces if many training pairs of (original, anonymized) faces are available. Contrarily, if the face recognition system is finetuned with a few examples of anonymized faces as an augmentation, may be it will start recognizing anonymized faces correctly. It is important to discuss the feasibility of such attacks.

4)  Similarly, in the case of identity recovery, it is not sufficient to show that reconstructed images have high perceptual similarity to the original images. It must shown that the reconstructed image matches with the original images with high probability using a face recognition system.

5) The diversity component of the proposed solution is not well-motivated. What is the application need for ensuring diversity? If the diversity loss is removed completely, will the same facial identity be generated every time irrespective of the input noise pattern (added to the initial latent code). If yes, is the model learning a one-to-one mapping between the latent code and generated image irrespective of the noise?

6) According to the introduction, "face images processed by visual identity information hiding methods are unrecognizable to human observers but can be recognized by machine". The images generated by the proposed method clearly shows the presence of a face image to a human observer. Only, the identity is partially hidden. In this case, how is this different from anonymization?

7) What is meant by "conditional embedding" of an image? What is the role of the "time modulation" module, which has been prominently highlighted in Figure 2, but never described in the text?

**Questions:**

Please see weaknesses.

---

> ### Author Response · Authors · 2023-11-16
> **Part 1 of Response to Reviewer jBnK**
>
> Thank you for your comments and feedback. We address your concerns here.
>
> >Q1: The paper is extremely hard to read and understand. A number of mathematical notations and terms (e.g., key-E, conditional embedding, etc.) are not defined clearly, which makes the proposed approach hard to follow.
>
> A1: We have made our best efforts to explain mathematical notations and terms to make our manuscript easy to read and understand. In the revised manuscript, we have furnished a comprehensive introduction to the definition of conditional embeddings and the method for acquiring them in Sections 2.1 and 2.2. In brief, conditional embedding is the learnable vector in stable diffusion model's text conditional space, which can help the diffusion model reconstruct a given image. We designed the MSI module to learn the conditional embedding of the given image in the text conditional space (Eq.6 and Eq.7).
>
> >Q2: First and foremost, the most basic requirement in a security/privacy paper is stating the threat model and assumptions explicitly. What information is being protected and from whom? What are the capabilities of the adversary? In this paper, no clear threat model has been presented. For example, the goals of anonymization and visual information hiding appear to be quite different. Is the proposed framework designed to meet both these objectives simultaneously? Or is the paper trying to propose a common framework that can achieve either one of these objectives at a time by slighting tweaking some parameter/loss?
>
> A2: Thanks for your suggestion. Our privacy protection method aims to safeguard personal identity information and prevent its collection and misuse by threat models, such as illegal snoopers, unauthorized automatic recognition models, and malicious facial manipulation models. As you mentioned in your comment, there are notable distinctions between the objectives of anonymization and visual identity information hiding tasks. Consequently, existing methods can only accomplish one of these goals. In this paper, we propose a unified framework that can achieve either anonymization or visual identity information hiding tasks by slightly tweaking some parameters during inference.
>
> >Q3: For anonymization (de-identification), it is not sufficient to show that "anonymized" faces cannot be recognized by a standard face recognition system. It is necessary to prove that the face cannot be "deanonymized" by a malicious adversary. In the proposed approach, it appears that complete recovery is possible provided the so-called "keys" are available. This makes the privacy dependent on the confidentiality of the "key". Even in the absence of the "key", it may be possible for an adversary to learn the inverse mapping between the original and anonymized faces if many training pairs of (original, anonymized) faces are available. Contrarily, if the face recognition system is finetuned with a few examples of anonymized faces as an augmentation, may be it will start recognizing anonymized faces correctly. It is important to discuss the feasibility of such attacks.
>
> A3: Potential attackers lack the incentive to recover the original image as they are unaware that it is encrypted. This is especially true when the encrypted image closely resembles a real face image. Even if malicious adversaries are aware of encryption, they can not access paired data (only have the anonymized version). Only in the event of our model being fully leaked can these adversaries obtain paired data. However, it is widely agreed among all anonymization methods that encryption models remain invisible to attackers.

---

> ### Author Response · Authors · 2023-11-16
> **Part 2 of Response to Reviewer jBnK**
>
> >Q4: Similarly, in the case of identity recovery, it is not sufficient to show that reconstructed images have high perceptual similarity to the original images. It must shown that the reconstructed image matches with the original images with high probability using a face recognition system.
>
> A4: We included the recognition rate of the recovered image in Table 1 of the previous manuscript, demonstrating that the recovered image generated by our method closely matches the original images when using the face recognition system. By speculating on the reviewer's intention, we have further calculated the perceptual similarity between the recovered image and the original image in the recognition model. Specifically, we input both images into the face recognition model to extract identity embedding and then calculate their Cosine similarity (Cos-IE). Cos-IE has higher precision than the recognition rate, which can better reflect the similarity between the recovered image and the original image in the embedding space of the recognition model. As shown in the table below, our method produces a high Cos-IE value compared to existing anonymization methods, indicating its superiority. In terms of visual identity information hiding task, our method achieves comparable performance in Cos-IE with AVIH.
>
> Quantitative comparison of recovered results. We compared our method with other anonymization methods (FIT, RIDDLE) and visual identity information hiding method (AVIH) in terms of image recovery performance. (F) and (A) represent that we use FaceNet and ArcFace as face recognition models, respectively.
>
> | Method     | FIT    | RIDDLE | Ours   | AVIH (F) | AVIH(A) | Ours   |
> |------------|--------|--------|---------|--------|----------|---------|
> | MSE        | 0.006  | 0.045   | **0.003**  | **0.003**    | 0.004  | 0.004  |
> | LPIPS      |0.051  | 0.192  | **0.037** | 0.216    | 0.109   | **0.059**  |
> | SSIM       | 0.762  | 0.494   | **0.854**  | 0.775    | 0.793   | **0.872**  |
> | PSNR       | 28.693 | 19.489 | **28.9**   | **32.306**   | 31.369   | 31.913 |
> | Cos-IE (F) | 0.896  | 0.774   | **0.956**  | 0.926    | 0.87   | **0.929**  |
> | Cos-IE (A) | 0.88   | 0.709 | **0.932**  | **0.919**    | 0.726   | 0.905  |
>
>
> >Q5: The diversity component of the proposed solution is not well-motivated. What is the application need for ensuring diversity? If the diversity loss is removed completely, will the same facial identity be generated every time irrespective of the input noise pattern (added to the initial latent code). If yes, is the model learning a one-to-one mapping between the latent code and generated image irrespective of the noise?
>
> A5: Diversity is crucial. If diversity is not emphasized, like some existing methods [1,2,3] do, the anonymous faces tend to look similar for different individuals. This makes it easy for an attacker to recognize that these faces are anonymized. Moreover, in certain situations, such as online conferences [4], the de-identified faces of participants should be distinct from one another. To ensure reliable identity protection, it is also important to consider variations in ethnicity, age, gender, and other facial features. Even after removing diversity loss (as shown in Fig.14), the model can still generate images with different identities because the noise added to the initial latent code varies. However, when a specific noisy latent code is used, the diversity of image identities generated decreases due to the identity dissimilarity loss guiding the anonymous image towards a subspace that is least similar to the original image.

---

> > ### Author Response · Authors · 2023-11-16
> > **Part 3 of Response to Reviewer jBnK**
> >
> > >Q6: According to the introduction, "face images processed by visual identity information hiding methods are unrecognizable to human observers but can be recognized by machine". The images generated by the proposed method clearly shows the presence of a face image to a human observer. Only, the identity is partially hidden. In this case, how is this different from anonymization?
> >
> > A6: The distinction between visual identity information hiding and anonymization lies in the varying targets of machine recognition. Anonymization aims to prevent machines from correctly identifying the person, whereas visual identity information hiding ensures accurate machine recognition. Previous methods of visual identity information hiding only focused on making images unrecognizable to humans while still being recognizable by machines. To enhance privacy protection security, we designed specific modules that ensure the generated image resemble a real face. Consequently, even if attackers obtain encrypted images, it is difficult for them to realize that the images are encrypted.
> >
> > >Q7: What is meant by "conditional embedding" of an image? What is the role of the "time modulation" module, which has been prominently highlighted in Figure 2, but never described in the text?
> >
> > A7: Thank you for the opportunity to clarify. In the revised manuscript, we have presented a thorough introduction to conditional embedding, the time modulation module, and other elements. In brief, conditional embedding is the learnable vector in stable diffusion model's text conditional space, which can help the diffusion model in reconstructing a given image. We designed the MSI module to learn the conditional embedding of the given image in the text conditional space (Eq.6 and Eq.7). The time modulation module integrates time information with the extracted features through a time embedding layer, adaptively adjusting the information intensity derived from the features and improving the quality of image generation.
> >
> > **Reference**
> >
> > [1] Gafni, Oran, Lior Wolf, and Yaniv Taigman. "Live face de-identification in video." Proceedings of the IEEE/CVF International Conference on Computer Vision. 2019.
> >
> > [2] Hukkelås, Håkon, Rudolf Mester, and Frank Lindseth. "Deepprivacy: A generative adversarial network for face anonymization." International symposium on visual computing. Cham: Springer International Publishing, 2019.
> >
> > [3] Wu, Yifan, Fan Yang, and Haibin Ling. "Privacy-protective-gan for face de-identification." arXiv preprint arXiv:1806.08906 (2018).
> >
> > [4] Maximov, Maxim, Ismail Elezi, and Laura Leal-Taixé. "Ciagan: Conditional identity anonymization generative adversarial networks." Proceedings of the IEEE/CVF conference on computer vision and pattern recognition. 2020.

---

> > > ### Author Response · Authors · 2023-11-22
> > > **Additional Response to Reviewer jBnK**
> > >
> > > Dear Reviewer jBnK,
> > >
> > > Thank you for your valuable feedback on our submission. We have read your comments carefully and have addressed them in our rebuttal. As the rebuttal process is ending soon, we would be grateful if you could acknowledge if our responses have addressed your comments. We would also be happy to engage in further discussions if needed. Thank you again for your time and consideration.

---

> > > > ### Comment · Reviewer_jBnK · 2023-11-22
> > > > **Response to Author Rebuttal**
> > > >
> > > > Thanks for the detailed response. I have gone through it in detail, but I'm still not convinced about the threat model and assumptions. Hence, I would like to retain the original rating.

---

### Author Response · Authors · 2023-11-16
**Summary Response**

We thank all reviewers for their questions and constructive feedback. We have undertaken substantial revisions to enhance the manuscript's readability. Key changes in the revised submission include:

1. We have incorporated a description of the threat model in Section 1, underscoring our privacy protection objectives.

2. We have elucidated the versatility of our proposed framework. Our proposed method can achieve anonymization or visual information hiding tasks by slightly tweaking some parameters during inference.

3. We have added annotations for key-E and key-I in Fig.2 and Fig.6.

4. We have furnished a more explicit definition of the three stages of our method in Section 2.

5. We have provided a more precise explanation for 'C' in Eq.2.

6. We have rewritten Section 2.2 to clearly define conditional embedding and elucidate the learning process.

7. In section 2.3.1, we have explained that unconditional embedding is the default embedding value when the condition is null.

8. Algorithm 1 has been revised to accurately depict the inference process of our method.

9. In the evaluation of image recovery, we have added the Cos-IE metric based on the reviewer's feedback. This metric measures the cosine similarity between the recovered image and the original image in the embedding space of the recognition model (as shown in Table 2).

10. We have furnished a comprehensive explanation of the symbols in Eq. 8 and added more details about the energy function in Appendix A.2.1.

11. In addition to the above modifications, we have addressed grammar issues and clarified statements throughout our manuscript.

We hope that these changes strengthen the state of our submission. Furthermore, we remain committed to ongoing revisions of our manuscript to enhance its readability and comprehensibility.

---

### Meta-Review · Area_Chair_3jvh · 2023-12-05

**Metareview:**

**Summary:**

This paper introduces a novel face privacy protection method using diffusion models, focused on unifying anonymization and visual identity information hiding. The core idea is to guide the denoising process of the diffusion model through an energy-based identity guidance module and learned conditional embeddings. The results demonstrate that the proposed method exhibits state-of-the-art performance in both tasks.

**Strengths:**

1. The paper presents a unique approach using diffusion models for facial privacy protection, marking a significant advancement in the field.
2. The authors thoroughly compare their work with existing literature, and the method is validated both qualitatively and quantitatively.
3. The paper is well-organized, making it accessible and understandable.

**Weaknesses:**
1. One reviewer has somer concern regarding the threat model and assumptions.
2. The diffusion-based method's inference efficiency is noted to be less than that of GAN-based methods.

PC/SAC Comment: After calibrating for systematically inflated reviews, a careful consideration was made on this paper.  The paper is indeed challenging to read, and a clear statement of goals and assumptions was lacking (as noted in the more critical reviews).  The authors brought up reasonable points during the discussion, but this paper could use a major revision to improve clarity.

**Justification For Why Not Higher Score:**

One reviewer suggests that the authors need to strengthen the description of the assumptions underlying the task.  The writing clarity could also be improved to more clearly convey these fundamental ideas.

**Justification For Why Not Lower Score:**

N/A

---

### Decision · Program_Chairs · 2024-01-16

Reject